# Understanding How School-Based Interventions Can Tackle LGBTQ+ Youth Mental Health Inequality: A Realist Approach

**DOI:** 10.3390/ijerph20054274

**Published:** 2023-02-28

**Authors:** Elizabeth McDermott, Alex Kaley, Eileen Kaner, Mark Limmer, Ruth McGovern, Felix McNulty, Rosie Nelson, Emma Geijer-Simpson, Liam Spencer

**Affiliations:** 1School of Social Policy, University of Birmingham, Birmingham B15 2TT, UK; 2School of Health and Social Care, University of Essex, Colchester CO4 3SQ, UK; 3Population Health Sciences Institute, Newcastle University, Newcastle upon Tyne NE2 4AX, UK; 4Faculty of Health and Medicine, Lancaster University, Lancaster LA1 4YW, UK; 5School of Sociology, Politics and International Studies, University of Bristol, Bristol BS8 1TU, UK

**Keywords:** mental health, LGBTQ+ youth, adolescence, sexual/gender minority

## Abstract

Globally, research indicates that LGBTQ+ young people have elevated rates of poor mental health in comparison with their cisgender heterosexual peers. The school environment is a major risk factor and is consistently associated with negative mental health outcomes for LGBTQ+ young people. The aim of this UK study was to develop a programme theory that explained how, why, for whom, and in what context school-based interventions prevent or reduce mental health problems in LGBTQ+ young people, through participation with key stakeholders. Online realist interviews were conducted in the UK with (1) LGBTQ+ young people aged between 13–18 years attending secondary schools (*N* = 10); (2) intervention practitioners (*N* = 9); and (3) school staff (*N* = 3). A realist retroductive data analysis strategy was employed to identify causal pathways across different interventions that improved mental health outcomes. The programme theory we produced explains how school-based interventions that directly tackle dominant cisgender and heterosexual norms can improve LGBTQ+ pupils’ mental health. We found that context factors such as a ‘whole-school approach’ and ‘collaborative leadership’ were crucial to the delivery of successful interventions. Our theory posits three causal pathways that might improve mental health: (1) interventions that promote LGBTQ+ visibility and facilitate usualising, school belonging, and recognition; (2) interventions for talking and support that develop safety and coping; and (3) interventions that address institutional school culture (staff training and inclusion polices) that foster school belonging, empowerment, recognition, and safety. Our theoretical model suggests that providing a school environment that affirms and usualises LGBTQ+ identities and promotes school safety and belonging can improve mental health outcomes for LGBTQ+ pupils.

## 1. Introduction

Young people who identify as lesbian, gay, bisexual, transgender, or queer/questioning (LGBTQ+) (We use LGBTQ+ to refer collectively to sexual minority and gender diverse identities because of the proliferation of terms used by young people. References to other research use the author’s original terminology for sexuality/gender.) experience significant mental health inequalities in comparison with their cisgender heterosexual (cis-hetero) peers [1]. International research consistently shows that LGBTQ+ young people have a higher prevalence of depression, self-harm, suicidality, and problematic substance use than cis-hetero young people [2,3]. Trans and gender variant youth are more likely to suffer discrimination in schools compared with cisgender youth [4,5], and a recent meta-analysis of studies comparing suicidality in youth has shown that trans youth were six times more likely to report a history of attempted suicide than cis-hetero youth, bisexual youth were five times more likely, and lesbian and gay youth were four times more likely [6].

The school environment (or climate) is a major risk factor and is consistently associated with negative mental health outcomes for LGBTQ+ young people [4,7,8]. The school climate includes all aspects of a young person’s school experience, such as the quality of teaching and learning, school community relationships, school organisation, and the institutional and structural features of the school [9,10]. UNESCO’s ‘Out in the Open’ global review reported that a significant proportion of LGBTQ+ young people experience school discrimination and/or violence based on their sexual orientation or gender identity or expression and that this can affect young people’s education, employment prospects, and mental wellbeing [11]. A UK-based study of over 7000 young people aged 16 to 25 found that the majority of young LGBTQ+ people believe their time at school is associated with hostility or fear, with consequences such as feeling excluded, achieving lower grades, and having to change schools. Most students reported that their school did not provide adequate support for their sexual orientation or gender identity [12]. A systematic review examining the consequences of non-inclusive sex education in schools for LGBTQ young people found that LGBTQ young people felt ill-equipped to navigate sexual health and relationships safely, and experienced mental health problems including suicidality. To ensure the emotional health and self-esteem of LGBTQ young people, it was recommended that the dominance of heteronormative relationship and sex education (RSE) must be reviewed [13]. UNESCO state that the education sector has a responsibility to provide safe and inclusive learning environments for all young people, and that effective education sector responses to discrimination based on sexual or gender identity requires a ‘comprehensive approach’ that involves the whole sector. However few countries currently have all the elements of a comprehensive education sector response in place [11].

Increasingly, research indicates that attempts to make the school climate more inclusive for LGBTQ+ pupils can improve education and mental health outcomes [4,14,15,16]. A recent systematic review identified key components of an LGBTQ+ positive school climate including: the presence of supportive staff; LGBTQ+ support groups; inclusive curricula; inclusive policies explicitly providing protections based on sexuality and gender identity; and staff training. The review found that an LGBTQ+ inclusive school climate reduced suicidality and depressive symptoms among LGBTQ+ young people [15]. Further, systematic reviews of research on homophobic bullying at schools have identified inclusive education policies and supportive curricula as factors which reduce homophobic bullying and lead to increased perceptions of school safety and more school belonging [17]. These inclusive education policies and supportive curricula can also serve as protective effects for the mental health of LGBTQ+ youth, reducing suicidal ideation and suicide attempts [18,19], particularly when the policies and curricula are well-established within a school setting (over three years) [20]. These policies promote positive safe interactions with others that improve feelings of belonging for LGBTQ+ youth [21]. School inclusion policies are particularly important in communities where LGBTQ+ people are marginalised, or where anti-LGBTQ+ stigma can influence policies [22].

Three systematic reviews have identified the importance of the school climate to trans and gender diverse youth mental health [4,5,23]. ‘School protective factors’ (a composite of teacher intervention, inclusive policy, availability of LGBTQ information, inclusive curricula, and presence of a gay-straight alliance (GSA) group) were associated with connection to an adult at school and feelings of safety [4]. These facilitated healthy relationships with peers and supportive, trusted school staff and were crucial in reducing poor mental health and promoting resilience amongst trans and gender variant students [5,23]. Martín-Castillo et al. emphasise the importance of relationships, with both teachers and peers, as essential to addressing and overcoming school victimisation, and fostering a sense of social acceptance and belonging [5]. The presence of a trusted adult, with whom young people can talk about personal issues, including gender and sexuality, helps them to feel that they are being seen and heard, can improve self-esteem [24], and reduce poor mental health outcomes [25].

Whilst recent systematic reviews do not consistently focus on or report the impact of the school climate on mental health outcomes, there are a number of school-based interventions that have been shown to improve mental health outcomes for LGBTQ+ youth [26]. For example, an empirical evaluation of a film-based intervention, ‘Out in Schools’, which was designed to reduce sexual orientation prejudice and foster inclusive school attitudes, found that the intervention was associated with reduced odds of LGB students experiencing discrimination, and both LGB and heterosexual female students being bullied or considering suicide [27]. School based LGBTQ+ support groups, such as gay-straight alliances (GSAs), have been shown to promote resilience for LGBTQ+ (and heterosexual) youth and improve mental health [28]. Research suggests that peer support is a critical component of school-based LGBTQ+ support groups because it can encourage members to develop a sense of community, advocacy, and provide support and friendship [29], which can enable members to validate and affirm their identities and expressions [30], develop greater self-confidence, self-esteem, and coping strategies [25,30,31], and lead to the reporting of lower levels of victimisation and suicide attempts [19]. A pilot study of a minority stress-informed mental health promotion program within the context of a GSA, found that those who attended the sessions reported them to be enjoyable, informative, relevant to their lives, and potentially helpful for other LGBTQ students [32]. A pilot study of a multi-ethnic sexual minority affirmative school-based group counselling intervention called ‘affirmative supportive safe and empowering talk’ (ASSET) found that self-esteem and proactive coping increased significantly across all ethnic subgroups, with the intervention showing promise in enhancing the resiliency of multi-ethnic sexual minority youth in school-based settings [31]. However, research examining GSAs, school functioning and mental health indicates that for LGBTQ+ students of colour, the association between the presence of a GSA and mental health and substance use is not as strong as it is for non-Hispanic white students [14]. In a review of LGBTQ student experiences in schools over a ten-year period in psychology journals, Abreu and colleagues report an over-representation of heterosexual voices (both students and adults) and corresponding under-representation of LGBTQ+ youth voices. They found that the majority of research had cisgender samples that were biased towards white young people and lacked critical reflection on the intersections of race and ethnicity with other identities [33].

At present the evidence tells us that poor school experiences such as homophobic, biphobic or transphobic bullying can have negative consequences for the mental health of LGBTQ+ youth. There is also a significant body of research, mainly from the USA and Canada, that indicates that improving the school climate for LGBTQ+ pupils through the presence of supportive staff, LGBTQ+ support groups, inclusive curricula and LGBTQ protective inclusive policies can improve mental health outcomes. However, there is a paucity of evidence [4,11] that examines the deliberate introduction of school-based interventions aimed at improving the mental health of LGBTQ youth. Most research is quantitative, utilizing already existing large-scale datasets, and examines the consequences of already existing improvements to the school climate. Consequently, qualitative, theory-driven, or mixed methods research that provides explanations for how and why interventions may improve the mental health of LGBTQ+ youth is limited. This article addresses this gap and reports on the second stage of a two-part study that utilized a realist method. In the first stage of the study, we produced a realist review of published evidence on school-based interventions to reduce mental health problems in LGBTQ+ young people (under review elsewhere) [34]. This review identified the positive interventions that supported LGBTQ+ mental health in school environments. Realist enquiry aims to open the ‘black box’ and theorise the program (interventions), unearthing the mechanisms (causal processes) which are triggered by the particular context to produce outcomes. The primary studies included in our review tended to focus upon outcomes and rarely detailed the underlying mechanistic processes, which is a common limitation of realist synthesis [35]. To address this limitation, the second stage of our study, reported in this paper, aimed to utilize empirical qualitative research to generate stakeholders (young people, school staff, intervention practitioners) perspectives and experiences to reflect on the programme theory that we developed and to answer the research question: how, why, for whom and in what context may school-based interventions prevent or reduce mental health problems in LGBTQ+ young people?

## 2. Materials and Methods

### 2.1. Realist Methodology

Realist methodology is founded on the ontological principle that social reality is interpretative and that social actors (humans) evaluate social reality [36] which means ‘*to understand how outcomes are generated, the role of external reality and human action need to be incorporated*’ [37]. A realist approach seeks to identify the underlying theories that explain patterns in how individuals make similar decisions under specific intervention conditions (demi-regularities). In the realist view, social programs (interventions) that seek to resolve or improve a complex social problem such as mental health, must understand what enables humans to change their feelings, thoughts, beliefs, or actions as a result of the ideas or opportunities introduced via the intervention. Interventions can initiate a whole range of potential mechanisms whose effectiveness will be modified in different contexts.

### 2.2. Realist Causation

The realist principle of generative causation is the key to knowing how and why an intervention works, the focus is the ‘mechanics of explanation’ or explaining ‘how things change’ [38]. The realist explanatory logic does not rely on the observation and control of reality that is usual in experimental design but views causal outcomes that follow from hidden underlying mechanisms acting in certain settings or contexts (Context–Mechanism–Outcomes, or CMOs). CMOs are theories, and configuring CMOs is the basis for generating and/or refining the programme theory that explains the underlying assumptions about how an intervention works. In other words, CMOs explain at a micro- and macro-level why the introduction of, for example, an LGBTQ+ curricula in schools, may potentially be seen by LGBTQ+ pupils as inclusion and generate a positive school connection and experience thus improving mental health.

Our realist review of published research produced a programme theory based upon synthesised literature examining school-based interventions that impact upon LGBTQ+ mental health [34]. However, much of the identified literature did not include the necessary detail to fully plot the causal pathways and detail the generative causal connection with the level of granularity required within realist methodology. To generate a more robust casual explanation for the conditions in which an intervention might improve mental health, we required empirical evidence gathered within a realist enquiry that could illuminate the different perspectives, action, and reactions that our programme theory suggests works. The aim of this stage of the study was to gain stakeholder perspectives on the underlying mechanisms that explain how, why, for whom, and in what context school-based interventions may work to improve the mental health of LGBTQ+ young people.

### 2.3. Young People’s Involvement

Two LGBTQ+ young people were involved as experts through experience to support the research. The young people were recruited prior to commencement of the project via local networks with LGBTQ+ youth organisations and were employed by the university as paid researchers. The LGBTQ+ young people took part in research meetings over a 12-month period. During these meetings, the LGBTQ+ young people helped to develop and refine the materials used for participant recruitment and data collection. They were also consulted on potential non-academic outputs of the research, and their advice and input were used to inform the project’s stakeholder engagement strategy.

### 2.4. Ethics

There are a number of ethical challenges that arise in asking young people to participate in a study related to LGBTQ+ issues, primarily relating to the risk of discrimination (due to being identified or perceived as LGBTQ+) or harm (e.g., emotional distress). We mitigated against these risks in the following ways: (a) provision of signposting to LGBTQ+ youth specific support services for participants; (b) identification of an agreed trusted adult at the participant’s youth group/service; and (c) a procedure for reporting risk and adverse events.

LGBTQ+ young people aged 13–18 were afforded the right to give consent without the need to involve their parents/guardians. Requiring parental consent may place the young person at risk of hostility, abuse, and rejection if their parents/carers were previously unaware of their sexual orientation or gender diversity [39]. It is increasingly recognized that young people under 16 years old are able, and should, give consent for taking part in research as long as they are judged as competent [40]. It is now common practice for research studies examining LGBTQ+ young people and health to waive parental consent [41]. The LGBTQ+ young people advising this study confirmed that it was important that parent/carer consent was waived to encourage young people to participate in the research. All decisions regarding consent were approved by the Faculty of Health and Medicine Research Ethics Committee at Lancaster University.

### 2.5. Sample Recruitment and Demographics

The target population for recruitment to the study were (1) young people aged between 13–18 years of age attending UK secondary schools who identified as LGBTQ+; (2) intervention practitioners e.g., those working within organisations who had delivered LGBTQ+ inclusivity interventions in UK schools; and (3) school staff, e.g., teachers and support staff. Participants were selected via a purposive sampling strategy [42], using contacts within national and local youth and LGBTQ+ organisations, to produce a diverse sample in terms of sexual orientation, gender identity, ethnicity, disability, and socioeconomic status. All participants recruited lived in England.

Recruitment took place during the outset of the COVID-19 pandemic and there were significant recruitment challenges due to the pressures placed upon the staff, management, and administration of the schools. To reduce additional pressure on schools during this time, young people were recruited through our contacts within youth and LGBTQ+ organisations, not through schools directly. In total, *N* = 22 stakeholder interviews were undertaken (see participants demographic characteristics in Table 1).

### 2.6. Data Collection

In line with the realist approach, we collected data within theory-driven interviews. Applying the teacher–learner cycle advocated by Pawson and Tilley [38], we explored the propositions generated from our realist review [34], refining, and developing our programme theories (CMOs) [38,43]. Realist interviews were conducted with key stakeholders between May and November 2021. These interviews were conducted online (using Zoom/Teams) with adults (intervention practitioners and school staff) and the interview guide covered the following core topics: (a) properties of anti-HBT and LGBTQ+ affirming interventions in schools, exploring the format, content and tone of interventions, the quality of delivery, attendance, and the impact of interventions; (b) intra-contextual variation and how this causally connects with; (c) the impact of these interventions on young people’s mental health.

Interviews using digital technology are becoming increasingly recognised as an accepted data collection method in qualitative research [44]. Text-based ‘instant messaging’ interviews on WhatsApp were conducted with LGBTQ+ young people because both previous studies and our LGBTQ+ youth advisory group indicated that this interview mode increases participation, facilitates discussion, and gives young people a greater sense of control and empowerment in the interview context [44]. The practical advantages of ‘instant messaging’ interviews include reduced cost, minimisation of travel, and improved access to participants who might not want, or be able, to take part in face-to-face interviews. However, some concerns have been raised regarding participants’ access to and familiarity with technology and ethical issues regarding data protection [45]. We worked with our LGBTQ+ youth advisory group, and Lancaster University’s Faculty of Health and Medicine Research Ethics Committee in order to overcome any potential risks in the present study. The WhatsApp interview guide was piloted with young people, and amendments were made to clarify the focus of the questions. Each question was prepared as an image file to be sent sequentially during a young person’s interview, and questions were intended to ‘test’ the programme theory we developed from our realist review and particularly how and why different interventions (e.g., inclusive school curriculum) may or may not improve mental health (see Appendix A). All participants were asked how likely it was that each of the eight intervention components would improve school climate and mental health, regardless of whether these interventions had been implemented in their respective settings.

### 2.7. Data Analysis

We utilized a realist iterative retroductive data analysis strategy to develop the programme theory [46,47,48]. The purpose of a retroduction reasoning technique is “*the identification of hidden causal forces that lie behind identified patterns or changes in those patterns”* [47]. Retroduction employs both inductive (drawing conclusions from the specific to the general) and deductive reasoning (drawing conclusions from the general to the specific) [49], as well as researcher insights, to comprehend generative causation by investigating the underlying social and psychological drivers identified as influencing outcomes [48]. Our analytic focus was on the context of the interventions and the development of causal pathways that were gaps in our original programme theory [34]. The analytical strategy consisted of three stages: (1) deductive and inductive coding; (2) identification of CMO configurations; (3) inductive thematic analysis.

In the first stage we conducted deductive and inductive coding. An initial pilot coding was carried out by two members of the research team applying a deductive coding frame (D-coding frame) that was based on the programme theory developed from our realist review [34], and simultaneously identifying inductive codes (I-codes) from the data. A data analysis workshop with five members of the research team developed and refined the definitions and operationalisation of the D-coding frame and the I-coding frame. Coding was then conducted on the full data set using the refined combined D and I-coding frame.

The second stage of analysis involved the identification of CMO configurations. A code was assigned to each individual CMO in the dataset. These were then aggregated through discussions amongst five members of the research team to identify patterns and dissimilarities. Thirdly, a thematic analysis of the inductive codes was conducted to examine in detail how context impacted the success of interventions in improving mental health outcomes. A second data analysis workshop with five research team members produced a refined programme theory that explained how, why, for whom, and in what context school-based interventions may work to improve the mental health of LGBTQ+ young people.

## 3. Results

### 3.1. Programme Theory

In Figure 1 below, we present the model of the programme theory (PT) developed from our original realist review [34], which illustrates the complex inter-relationships between context, multilevel mechanisms, and outcomes. In our PT model we have three levels of mechanisms to capture the multiple causal pathways at which a mental health intervention may work e.g., psychological, behavioural, emotional, cultural, social. We aimed to make explicit and theorise how the intervention resources (mechanism 1, the orange layer in Figure 1) makes possible opportunities for different human change i.e., ‘what changes’ e.g., positive relationships (mechanism 2, the green layer in Figure 1) and ‘how it changes’ (mechanism 3, the blue layer in Figure 1) in terms of individual cognitive processes e.g., empowerment. It is important to note that the model is dynamic and not static. The arrows in the model are a recognition of the mutually reinforcing and overlapping relationship between context and the mechanisms.

In the subsequent sections we first discuss the contextual factors (red outer ring of Figure 1) that were crucial to the success of the interventions. We then present three broad causal pathways (CMOs) that explain why and how interventions may improve mental health.

### 3.2. Context Factors

In the programme theory model (Figure 1), the outer layer (red) contains the five context factors that our analysis at this stage of the study suggests are essential for school-based interventions to effectively improve mental health outcomes: a whole-school approach; intersectionality; collaborative leadership; school culture; and the legal, political, economic, and discursive environment. We discuss participant perspectives on each of these below.

#### 3.2.1. Whole School Approach

Intervention and school practitioners emphasized that the ‘whole school approach’ (where all parts of the school, including senior leaders, teachers and all school staff, as well as parents, carers, and the wider community, work together) meant that interventions had to have committed, long-term and embedded approaches to be successful and improve LGBTQ+ pupils’ mental health. By contrast, ambivalent, tokenistic and ‘add-on’ approaches were prominent explanations given for the partial or total failure of interventions. For example, ‘tacking’ LGBTQ+ topics onto the end of classes rather than embedding them throughout the curriculum, or introducing policies with no comprehensive plan for implementation, monitoring and review. As the LGBTQ+ young person quotation below illustrates, tokenism was not effective:

“We had LGBTQ displays, maybe one pride assembly a year, and an LGBTQ club, but little to none of it helped with understanding, much [of] those displays only really felt like they were doing it to help their image, instead of just doing it out of pure support and the support groups, since the location was plastered around, the homophobic guys tended to lurk near the room and just point and laugh, really.”(LGBTQ+ young person)

Participants stressed that the whole school approach encompassed pupils and staff at all levels, with a commitment to consistent intervention work, accompanied by evaluation and review processes, and attention to all aspects falling within the remit of school governance.

#### 3.2.2. Collaborative Leadership

Participants emphasized that the success of intervention was more likely if there was collaborative leadership between adults and pupils within the school. School senior leadership teams (SLTs) were referenced frequently as both an enabler of and barrier to the success of interventions, particularly in relation to establishing and driving school values, setting priorities, and allocating resources. In addition, participants stated that successful interventions needed a designated individual who facilitated the program:

“I went to the Head, and I said “look, this is something that I want to do a lot more of in school in terms of this” and she just gave me absolute carte blanche which has been brilliant. […] I think the really, really strong thing is that there’s one person that leads it. It’s like that go to.”(School Staff)

While the benefits of a key ‘go-to’ individual were discussed, SLT support was identified as essential for the potential of a designated individual acting as a go-to and change-driver within the school to be realised. The risks and limitations inherent in the over-reliance on a single individual’s passion and investment was a recurring theme. Our analysis at this stage of the study suggests that where adult leadership can be coupled with pupils’ leadership, the programmes had more success. Ideally, collaborative leadership recognises the forms of input that are appropriate and possible across different groups of people within a school and brings these together. However, intervention practitioners made reference to the risks of leaning too heavily on leadership by young people where this was not adequately supported and facilitated.

#### 3.2.3. Intersectionality

Intersectional approaches to intervention work, which included focusing on ethnicity, religion, disability, or other connecting identities, were mainly absent. Young participants viewed the lack of intersectional approach to LGBTQ+ school interventions as problematic, and wanted their whole identity to be recognised:

“I need celebrity representation of LGBT people from BME faith communities to make me feel empowered. My community and culture is part of my identity—I have not lost that.”(LGBTQ+ young person)

Adult participants argued that fixed and static identity approaches produced narrow homogenizing interventions that did not address young people experiencing multiple forms of marginalisation e.g., ethnicity and gender diversity. Participants described the importance of intervention approaches that understand mental health in terms of compound marginalisation and feelings of isolation, rejection, stress, and resentment:

“We know that LGBTQ young people have experienced poorer mental health than their non-LGBTQ peers, but that is more so for those who access free school meals, for those who are Black, for those who are disabled. It is suggested that the multiple experiences of isolation and knowing that the world is still a place that discriminates, then when the world is changing through stuff like COVID, the fear must be magnified, mustn’t it? So, I think doubly impacted by isolation plus anxiety about the state of the world.”(Intervention Practitioner)

#### 3.2.4. Legal, Policy, Economic and Discursive Factors

Financial resources were cited frequently as a challenge in the successful implementation of intervention work, in paying for training and consultation with third sector organisations for example, as well as paying teachers for leading on intervention work within a school. Staff time and capacity in a context of high demand and often competing pressures and priorities was also a major factor. The impact of COVID-19 had exacerbated these pressures considerably, leading in some cases to school LGBTQ+ groups folding, and intervention work being deprioritised:

“COVID has seen that actually schools haven’t prioritised these spaces, so therefore actually now we don’t really have focus on HBT bullying and or equally necessarily a commitment to creating LGBTQ+ safe spaces. Actually, it feels like regression, in that sense.”(Intervention Practitioner)

A lack of clarity and support around national legal and policy directives was also identified as a challenge. UK Guidance issued by the Department for Education and the Equalities and Human Rights Commission was characterised by participants as ambiguous and led to confusion and a lack of confidence or, in some cases, to ambiguities being exploited in order to delay or block intervention work, particularly in relation to young trans and gender diverse people:

“Some of the responses that we have seen, in the last two years, I guess, to trans inclusive initiatives have been reminiscent of the run up to Section 28 back in 1988, absolutely. And so I think the fear of some kind of replies from parents or the community is greater than what actually does happen when they actually do something.”(Intervention Practitioner)

A prominent factor contributing to these challenges was mainstream media and public discourse. In particular, intervention and school practitioners referenced the impact of anti-trans media rhetoric on schools in terms of the perpetuation of hostile attitudes to gender diversity among adult staff members, and as a factor contributing to school wariness about being seen to be engaging in proactive LGBTQ+ intervention work. In addition, divisive and polarising media rhetoric was described as a source of heightened anxiety and wariness, particularly around the question of faith.

These contextual factors (the red outer ring of Figure 1) were crucial to the success of the interventions, and we now present the three broad causal pathways (CMOs) that explain why and how interventions may improve mental health.

#### 3.2.5. Causal Pathway 1: Interventions That Promote LGBTQ+ Visibility

In our findings, interventions that made LGBTQ+ identities and communities visible such as affirmative displays (e.g., LGBTQ+ posters, notice boards, door stickers) curriculum inclusion (e.g., explicit reference to LGBTQ+ diversity among historical figures, inclusive sex and relationships education) and standalone input (one-off LGBTQ+ events and activities e.g., themed assemblies, LGBTQ diversity workshops, school-wide ‘Pride’ celebrations) (mechanism 1) seemed to impact most strongly on how young people thought of and felt about their own identities, as well as how they thought others perceived these identities (mechanism 2). This change took the form of young people feeling recognised and affirmed, feeling included and as though they belonged, and not feeling alienated or ‘different’ in ways they found distressing, thus usualising LGBTQ+ identities (mechanism 3) (see Figure 2). As one pupil stated, ‘curriculums being inclusive of LGBTQ folks are also really important to me as representation helps people to feel like they’re not alone’. The experience of belonging was expressed prominently in terms of not feeling alone and this was often described as feeling ‘less different’ or feeling ‘normal’ in relation to peers, and not feeling alienated from the self, and encouraging positive mental health.

#### 3.2.6. Causal Pathway 2: Interventions for Talking and Support

Our findings suggest that interventions that provided opportunities for LGBTQ+ young people to talk and get support such as external signposting (i.e., referral to services, organisations and groups outside of the school, e.g., local LGBTQ+ youth groups) and having a trusted LGBTQ+ inclusive adult available in school to talk about LGBTQ+ issues and not be misunderstood or problematised (e.g., counsellors, LGBTQ+ ‘champion’ (demonstrating leadership in equality, diversity and inclusion), teacher) (mechanism 1) impacted most strongly upon the quality of young peoples’ relationships within the school, their emotions and feelings, and their thoughts about themselves (mechanism 2) (see Figure 3). The following quotation from an LGBTQ+ young person interviewee explains:

“I need to speak to someone like a teacher. Who can understand me and offer support. My mates in school I would not speak to as they can judge quickly. Signposting is good but if you’re in a bad place it’s too much Information. Counselling would be good but guided and supported. It hard being yourself, I come from a religious family and my community does not accept of LGBT equality at all. I think mentally the struggles can be hard no doubt.”

The positive mental health impact of these interventions to promote school relationship networks included friendships and relationships with peers, as well as relationships based on trust and mutual respect with adults in the school. Young people frequently aligned being able to talk and having support with descriptions of feeling more safe and able to cope in school (mechanism 3), leading to better mental health.

#### 3.2.7. Causal Pathway 3: Interventions That Change Institutional School Culture

In our findings, ‘behind-the-scenes’ interventions such as LGBTQ+ staff training that developed competence and confidence across staff regarding understanding LGBTQ+ identities and experiences, (e.g., inclusive language, challenging HBT language and bullying); and the development, implementation, and ongoing review of explicit policies addressing LGBTQ+ diversity of a school student body (e.g., anti-HBT bullying, pronoun use) (mechanism 1), although less easily observed by young people, were most strongly allied with diffuse change across the school culture and ethos, and as manifest in the behaviours and actions between individuals in the school, both between students and between students and staff (Mechanism 2) (see Figure 4).

“Inclusion policies is very important, having policies that consider is a long-term strategy [...] the LGBTQ+ people in the school won’t feel shy/alone in the school. We would feel recognized and even empowered. Issues like bullying won’t happen anymore.”(LGBTQ+ young person)

As this young person suggests, explicit policies addressing LGBTQ+ diversity and staff training were clearly connected in the data to feeling safe from bullying and violence, both from other students and from teachers, as well as belonging and empowerment (mechanism 3), and a sense of confidence in adults at school for support and understanding, thus improving pupils’ mental health.

## 4. Discussion

The aim of this stage of the study was to go beyond an ‘it works’ description and to develop a theoretical understanding of why, how, for whom and in what context school-based interventions can improve the mental health of LGBTQ+ pupils. International research demonstrates that improving aspects of the school climate to provide an LGBTQ+ inclusive environment can promote LGBTQ+ pupils’ mental health. However, most of the evidence is quantitative and draws on already existing large-scale survey datasets. What is missing from the evidence are explanations for how interventions may work and qualitative research that provides the perspectives of all key stakeholders but especially those of LGBTQ+ young people.

It is clear from our data analysis that school-based interventions attempting to improve LGBTQ+ pupils’ mental health must directly tackle the dominant cisgender and heterosexual normative school environment, and that this must happen at multiple levels. Our programme theory posits that the ideal situation is where interventions to improve the school climate for LGBTQ+ young people work across the entire school system (whole-school approach) through committed, long-term, and embedded approaches including inclusive policies, staff training, curricula inclusivity, LGBTQ+ visibility and support groups. A whole-school approach goes beyond learning and teaching in the classroom, to pervade all aspects of school life. This includes the promotion of health and wellbeing, in which all members of the school community are committed to working collectively and collaboratively [50]. This should have a collaborative leadership approach between pupils and school staff with support from the school senior leadership team. To ensure that the mental health of all LGBTQ+ young people improves, the whole-school approach must be intersectional, i.e., with an understanding that LGBTQ+ identities are not fixed, and that young people may have multiple identities that inform their experience of school. For example, white LGBTQ+ pupils will not encounter racism and the discriminatory stereotypes that LGBTQ+ youth of colour endure.

Our findings also indicate that the legal, policy, economic and discursive context in which schools operate is crucial to the delivery of effective interventions to improve the LGBTQ+ experience in schools and therefore the mental health of LGBTQ+ students. The ideal context is one where there are clear national legal and policy directives to prioritise and address the poor experience of LGBTQ+ young people in schools. This then incentivises LGBTQ+ inclusivity work and enables schools to allocate the financial resources to pay staff and external organisations (for training and consultation) to implement the interventions. Ideally, the wider mainstream media and public discourse would be encouraging of LGBTQ+ equality, especially trans equality, and upholding of the rights of LGBTQ+ young people (particularly those who are gender diverse) to have discriminatory-free schooling, a positive education and good mental health. This would mean schools and school staff would feel confident to engage in proactive LGBTQ+ intervention work, particularly around the question of faith and gender diversity. Unfortunately, in most cases, the contemporary environment is not ideal. School environments are complex institutions with established social norms and processes, they are sites of intense scrutiny, intended especially to produce the next generation of citizens (young people). For decades in the UK, young people’s sexuality and gender has been a battleground within schools—e.g., Section 28, Relationship and Sex Education (RSE), Equality Act 2010 [51]—and this historical legacy is the social context in which contemporary interventions are introduced. As Pawson and Tilley [38] remind us, programs carry a ‘history’ and will only work if they introduce ideas and opportunities to groups/people if the social and cultural conditions are amenable to change.

In addition to the importance of context, understanding how interventions work was a key question for this study. Interventions can ‘fire’ a whole range of potential mechanisms, and we were interested in the micro- and macro-causal mechanisms that underly people’s choices, capacities, and feelings [38]. Realist mechanisms are theories that attempt to make explicit those causal processes that cannot be observed. The explanation of interventions that improve mental health must examine causation as one that includes the ‘liabilities, powers, and potentialities’ [38] of subjects and as acting internally and externally for these subjects. Separating context from mechanism is frequently experienced as challenging in realist enquiries [35,38,52] and for this reason, our programme theory had three layers of ‘change mechanisms’ [38] wherein we attempted to capture the interplay between the intervention resources e.g., staff training (mechanism 1), what changes e.g., pupil relationships (mechanism 2) and individual cognitive processes e.g., empowerment (mechanism 3). We suggest three main causal pathways that explain the improvement of LGBTQ+ mental health as a result of the interventions. Firstly, those interventions that promote LGBTQ+ visibility and facilitate the usualising of LGBTQ+ identities, school belonging, and recognition; secondly, interventions for talking and support that develop safety and coping; and thirdly, interventions that address institutional school culture (staff training and inclusion polices) that foster school belonging, empowerment, recognition, and safety.

These causal pathways are theories and a starting point, and we need much more research to develop our understanding of how school interventions work to improve school climate and the mental health of LGBTQ+ young people. A limitation of this study is that we were only able to recruit a small number of school staff, with the study conducted online during the height of the COVID-19 pandemic when school staff were operating in very difficult conditions. However, in realist methodology the unit of analysis is not the person, but the events and processes around them, so that every respondent can uncover a collection of micro events and processes, each of which can be explored in multiple ways to test theories [48]. A further limitation is that we were unable to examine aspects of individual school culture. Our original intention was to utilize a case study methodology that would have enabled a robust examination of school culture. Future research should conduct a realist evaluation of an intervention as it is implemented. However, in the UK, this is difficult given that there is no national directive to address LGBTQ+ inclusivity in schools.

## 5. Conclusions

There is global concern about the mental health of young people and the COVID-19 pandemic has both exacerbated the problem and heightened awareness of the importance of good schooling for positive pupil mental health [53]. The reluctance of nations worldwide in taking action in schools to tackle LGBTQ+ pupils’ marginalisation and inequality is a major injustice that has been highlighted by the UN [11]. The 2016 UN Convention for the Rights of Child Committee General Comment no.20 [54] on the implementation of child rights during adolescence specifically emphasizes the way that nation states should take effective action to protect lesbian, gay, bisexual, transgender and intersex young people from all forms of violence, discrimination or bullying and to improve mental health. Our findings suggest that providing a school environment that affirms and usualises LGBTQ+ identities and that promotes school safety and belonging can improve mental health outcomes for LGBTQ+ pupils. We now need the UK and other countries to take seriously LGBTQ+ young people’s rights and ensure they are afforded equal respect and protection as their peers in schools. We may then find that the mental health of LGBTQ+ young people improves.

## Figures and Tables

**Figure 1 ijerph-20-04274-f001:**
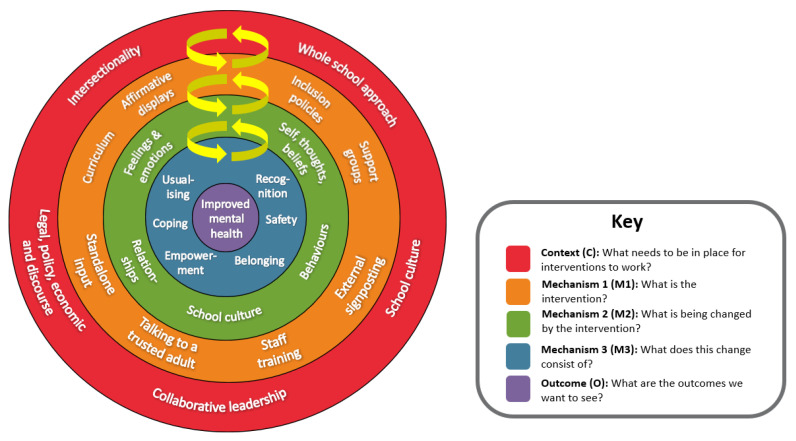
Programme theory.

**Figure 2 ijerph-20-04274-f002:**
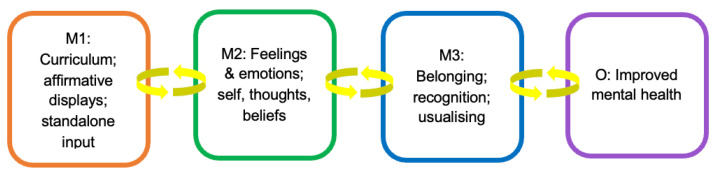
Interventions that promote LGBTQ+ visibility.

**Figure 3 ijerph-20-04274-f003:**
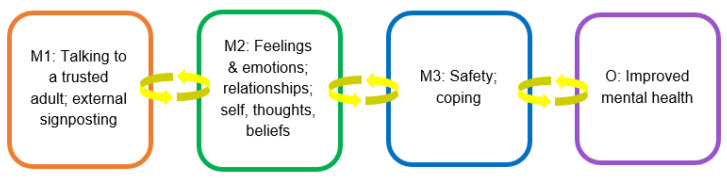
Interventions for talking and support.

**Figure 4 ijerph-20-04274-f004:**
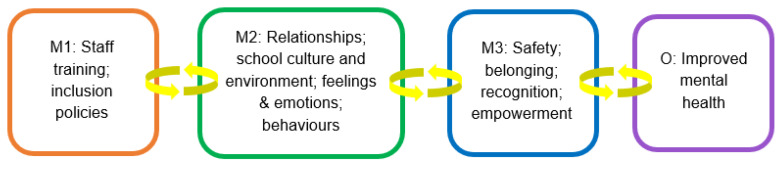
Interventions that change institutional school culture.

**Table 1 ijerph-20-04274-t001:** Participant demographics.

Variable	Classification	Young People (*N* = 10)	Intervention Practitioners (*N* = 9)	School Staff (*N* = 3)
Age	13–16	4	-	-
	17–18	6	-	-
	21–30	-	1	2
	31–40	-	5	-
	41–50	-	2	1
	51–60	-	1	-
Gender	Male	5	3	-
	Female	1	4	2
	Non-binary	3	1	1
	Other	1	1	-
Are you trans?	Yes	4	2	-
	No	4	7	3
	Unsure	2	-	-
Ethnicity	English/Welsh/Scottish/Northern Irish/British	7	7	2
	White (Other)	-	1	-
	European	-	1	-
	African	2	-	-
	Pakistani	1	-	-
	Jewish European	-	-	1
Sexual orientation	Lesbian	2	-	1
	Bisexual	3	1	-
	Gay	3	3	-
	Pansexual	1	-	-
	Queer	-	3	1
	Heterosexual	-	1	1
	Other	1	1	-
Education level	No qualifications	4	-	-
	GCSE	4	-	-
	AS Levels	1	-	-
	A Levels	1	-	-
	First Degree	-	6	2
	Higher Degree	-	3	1
Occupation	Student	8	-	-
	Unemployed	2	-	-
	Full-time Employment	-	6	2
	Part-time Employment	-	3	1
Do you have a disability?	No	7	6	2
	Yes	3	3	1

## Data Availability

Data available upon reasonable request.

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
