# Peer review of "Understanding How School-Based Interventions Can Tackle LGBTQ+ Youth Mental Health Inequality: A Realist Approach"

_ijerph, 2023, doi:10.3390/ijerph20054274_

Round 1
Reviewer 1 Report
There is one sentence in this article that provides the overarching context of the research. This is ‘The reluctance of nations worldwide in taking action in schools to tackle LGBTQ+ pupils marginalization and inequality is a major injustice that is highlighted by the UN [10].’. The combination of the literature and empirical data collection for this analysis shows a great depth of research into aims of the paper. The analysis of this data including participant voices provides an in-depth understanding of the LGBTQI+ discrimination and marginalization issues within the school environment. The authors have undertaken a comprehensive review of the data and the interpretative analysis provides several action options to counter the issue of the mental health of LGBQTI+ youth which are illustrated diagrammatically as flow charts and a relational contextual factors diagram.
The research includes the theoretical application of the ‘whole of school’ approach to inclusivity and understanding of LGBQTI+ youth and the emotional damage that can arise from a non-supportive environment. The authors relate the cause of these issues not only to the individual schools, but also to the national education funding to allow for the training and employment of suitable people to deliver the learning necessary to alleviate this discrimination. The authors have developed their theoretical approach to a solution with a realist iterative reproductive analysis that identifies the subjective experiences of the LGBTQI+ school cohort which offers an identified pathway for future development of their theoretical approach.
This is an insightful research paper for policy makers, teachers, support workers and others in this field of society. The paper as submitted has clarity of information and presentation for which the authors are commended.
Author Response
Many thanks for your encouraging comments regarding our submission. We have re-worked the manuscript to ensure written clarity, and address issues raised by the other reviewers. Please see attached revised manuscript.

Reviewer 2 Report
A bigger sample should be used for getting to such definitive conclusions. Although the method chosen is adequate, more participants and, above all, more school staff and practitioners are needed.
A deeper analysis of the theoretical background is also needed. Key materials related to the topic are missing.
Author Response
Thank you for your comments regarding our manuscript. In realist methodology the unit of analysis is not the person, but the events and processes around them, every respondent (every staff member, every young person) can uncover a collection of micro events and processes, each of which can be explored in multiple ways to test theories. This is in accordance with the RAMESES II guidance, and we have included reference to this in the text to ensure clarity (at 4. Discussion). We have noted in the limitations section of the study that a greater number of school staff participants would improve the results.
We believe, as do the other reviewers, that we have provided a thorough theoretical background to the study. Please see attached revised manuscript.

Reviewer 3 Report
The aim of this realist study was to develop a programme theory that explains how, why, for whom, and in what context school-based interventions prevent or reduce mental health problems in LGBTQ+ youth. This is an important and well-planned study, which results explain how context factors are critical to the delivery of successful interventions. The key and well-justified conclusion of the authors is that providing a school environment that affirms and usualises LGBTQ+ identities and promotes school safety and belonging can improve mental health outcomes for LGBTQ+ pupils. Such a practical conclusion may result in better outcomes of the future, very much-needed, interventions in this area as well as in (equally needed) legal and formal solutions.
The manuscript is clear, relevant to the field and presented in a well-structured manner. The cited references are mostly recent (last 10-15 years, which is in my opinion recent enough in this specific field) and the use of older sources (e.g. 38) is well justified. The study design / methodology is not well know but appropriate to test the hypothesis. However, as it is not a popular methodology, I suggest a few additional explanations. To increase the chances of the manuscript's results reproducibility I suggest adding a Table or Appendix presenting interview(s) questions (image files?) [p.7]. I believe this would help the readers, especially those not familiar with realist interviews, understand the used methodology much better. The figures and the table are clear and easy to understand.
The conclusions are consistent with the presented evidence. Ethics statements and data availability statements are adequate.
Specific comments:
Introduction: very thorough and well prepared. Page 3, the first paragraph is a bit of a repetition. You may consider rereading the introduction part and checking for possible repetitions.
Page 4, first paragraph - you write about "the first stage of the study" and "the second stage of the study". If I understand correctly, "the first stage was described previously in a different publication and "the second stage" is what this manuscript presents. I suggest rephrasing this paragraph to make it clearer to the reader what you address as an "our study" (this paper or a bigger project?).
Methodology
As mentioned previously, I suggest adding a table or Appendix presenting (core?) interview questions (at least for the text-based interviews with pupils).
Sample [or in Limitations of the study]
I suggest you justify why such a small sample is sufficient for the realist approach [for those unfamiliar with the approach, the N will seem far to small for a scientifically sound study], e.g. " Because the unit of analysis is not the person, but the events and processes around them, every respondent (every staff member, every participant) can uncover a collection of micro events and processes, each of which can be explored in multiple ways to test theories" (http://www.ramesesproject.org/media/RAMESES_II_Realist_interviewing.pdf).
Data analysis
Please explain better the "deductive coding" and "inductive coding".
Please explain how codes were aggregated to identify patterns and dissimilarities (by hand? by software? by whom? how many judges?).
Results
p. 8. point 3.2. and p. 11 point 3.2.2.- you write "our analysis suggests / our findings indicate". Is that about the analysis in this paper or in the previous paper? This is unclear (please read the text attentively to check the context of the use of this phrase).
p. 10, last citation of the Intervention Practitioner "... And so I think the fear of some kind of some kind...." [one "of some kind" is unnecessary]
p. 11 point 3.2.2. - who do you understand by LGBTQ_ 'champion' teachers?
At the end of some citations in the results section, there is a dot ".", while after others there is none. Please keep it coherent.
Discussion
p. 13 - I suggest avoiding such phrases as "it was crystal clear" in the paper. However, I am not an English native speaker, so feel free to ignore this comment.
Author Response
Many thanks for your encouraging and incredibly helpful point-by-point comments.
To address your concerns, we have:
- Attached Supplementary Material, which includes the Young People interview schedule used, to ensure transparency of data collection methods and demonstrate how this methodology was used.
- We have edited the Introduction section, to ensure there is no repetition.
- We have clarified the difference between the two stages of our project, and which is being reported on in this paper.
- Clarification of deductive and inductive reasoning included.
- We have aimed to improve clarity, and address formatting issues in the Materials and Methods and Results sections.
- In the Discussion section, we have provided a defence of the sample in accordance with RAMESES II guidance.
Please see attached revised manuscript.

Reviewer 4 Report
Thank you for sharing such an important piece. The literature review was very in-depth and thought out. Author states that parental consent was not taken for the LGBTQ+ youth that were interviewed. Author should elaborate that they received IRB or HSC approval to do this. The author should make it clear if the sample came from the same school or different schools. Also, what type of intervention these schools already have implemented for LGBTQ+ inclusivity. The author does not detail what type of interventions were implemented. The sample size for school staff was a very small event for a qualitative manuscript. It is hard to believe that saturation was met with a small sample and that these school staff members came from the same school or different schools. Again, the flaws are with the methods section and needs to be expanded on, so readers understand what the study was actually was.
Author Response
Many thanks for your encouraging comments. We have aimed to address your concerns in our revised manuscript, particularly around our Materials and Methods (consent and recruitment), and justification of sample size in the context of realist methodology (in Discussion section). Please see attached revised manuscript.

Round 2
Reviewer 2 Report
Thanks for taking comments into consideration.
Author Response
Thank you for positive response.
Reviewer 4 Report
Thank you for addressing the concerns from the reviewers and explaining the sample size.
Author Response
Thank you for positive response.